# Prognostic Role of Malnutrition Diagnosed by Bioelectrical Impedance Vector Analysis in Older Adults Hospitalized with COVID-19 Pneumonia: A Prospective Study

**DOI:** 10.3390/nu13114085

**Published:** 2021-11-15

**Authors:** Andrea Da Porto, Carlo Tascini, Maddalena Peghin, Emanuela Sozio, Gianluca Colussi, Viviana Casarsa, Luca Bulfone, Elena Graziano, Chiara De Carlo, Cristiana Catena, Leonardo A. Sechi

**Affiliations:** 1Division of Internal Medicine, Department of Medicine, University of Udine, 33100 Udine, Italy; gianluca.colussi@uniud.it (G.C.); viviana.casarsa@gmail.com (V.C.); luca.bulfone@gmail.com (L.B.); Cristiana.catena@uniud.it (C.C.); leonardo.sechi@uniud.it (L.A.S.); 2Division of Infectious Diseases, Department of Medicine, University of Udine, 33100 Udine, Italy; carlo.tascini@uniud.it (C.T.); maddalena.peghin@uniud.it (M.P.); emanuela.sozio@uniud.it (E.S.); Elena.Graziano@gmail.com (E.G.); chiaradecarlo@gmail.com (C.D.C.)

**Keywords:** malnutrition, COVID-19, BIVA, bioelectrical impedance vector analysis, low cellular mass

## Abstract

Background: Little is known on the clinical relevance of the nutritional status and body composition of patients hospitalized with SARS-CoV-2 infection. The aim of our study was to assess the prevalence of malnutrition in patients with COVID-19 pneumonia using bioelectrical impedance vector analysis (BIVA), and to evaluate the relationship of their nutritional status with the severity and outcome of disease. Methods: Among 150 consecutive patients who were hospitalized with COVID-19 pneumonia, 37 (24.3%) were classified as malnourished by BIVA, and were followed-up for 60 days from admission. Outcome measures were differences in the need for invasive mechanical ventilation, in-hospital mortality, and the duration of hospital stay in survivors. Results: During 60 days of follow-up, 10 (27%) malnourished patients and 13 (12%) non-malnourished patients required invasive mechanical ventilation (*p* = 0.023), and 13 (35%) malnourished patients and 9 (8%) non-malnourished patients died (*p* < 0.001). The average duration of the hospital stay in survivors was longer in patients with malnutrition (18.2 ± 15.7 vs. 13.2 ± 14.8 days, *p* < 0.001). In survival analyses, mechanical ventilation free (log-rank 7.887, *p* = 0.050) and overall (log-rank 17.886, *p* < 0.001) survival were significantly longer in non-malnourished than malnourished patients. The Cox proportional ratio showed that malnutrition was associated with an increased risk of mechanical ventilation (HR 4.375, *p* = 0.004) and death (HR 4.478, *p* = 0.004) after adjusting for major confounders such as age, sex, and BMI. Conclusions: Malnutrition diagnosed with BIVA was associated with worse outcomes in hospitalized patients with COVID-19 pneumonia.

## 1. Introduction

The pandemic caused by severe acute respiratory syndrome-Coronavirus-2 (SARS-CoV-2) has resulted in a heavy challenge for healthcare systems worldwide, leaving behind thousands of deaths. The extraordinary transmissional ability of SARS-CoV-2 has resulted in a quick spread of a disease that has been named COVID-19. It is therefore essential to shed light on patients’ features that could predispose to a more severe disease course. Older age, male sex, glycemic control, obesity, hypertension, chronic pulmonary disease, and malignances have been identified as strong clinical predictors of poor outcomes in hospitalized patients, as well as many biochemical biomarkers [1]. Malnutrition can be defined as “a state resulting from lack of intake or uptake of nutrition that leads to altered body composition (decreased fat free mass) and body cell mass leading to diminished physical and mental function and impaired clinical outcome from disease” [2]. Malnutrition can result from starvation, critical illness, or ageing (e.g., >80 years), alone or in combination [3]. Malnutrition, indeed, is common in hospitalized patients, with a prevalence between 20% and 50% [4], and is often associated with increased morbidity, mortality, and medical costs [5], most significantly in the critically ill population, where it is even more prevalent [6]. Despite the relevance of nutritional status and its prognostic impact for hospitalized patients, very little is known on the nutritional status and body composition of patients with COVID-19, its relationships with the severity of disease, and, most importantly, its prognostic role in hospitalized patients with critical disease.

Patients with COVID-19 are remarkably exposed to the risk of malnutrition. The intense inflammatory response caused by infection leads to a catabolic condition with increased nutritional requirements, while the need of huge amounts of oxygen delivered with facial masks or noninvasive ventilation makes regular feeding very challenging in these patients. To date, gold standards for the assessment of nutritional risk among vulnerable individuals and those who contract COVID-19 are lacking [7]. Current criteria for the diagnosis of impaired nutritional status (actual malnutrition) include and emphasize deleterious changes in body composition that are not considered within the standard questionnaires for assessment of risk of malnutrition [8]. Bioimpedance analysis (BIA) methods provide a practical bedside approach to the diagnosis of malnutrition [9] in the hospital setting.

The BIA technique measures whole body impedance, that is the opposition of the body to an alternating current consisting of two components: resistance (R) and reactance (Xc). Resistance indicates the fall of voltage caused by conduction through ionic solutions. Reactance indicates the delay of the current flow measured as the phase-shift reflecting dielectric properties of cell membranes and tissue interfaces. The most used and clinically relevant impedance parameter obtained by BIA is the phase angle. In past years, measurement of the phase angle has been applied to many clinical settings, with evidence of good reliability as a marker of nutritional status and a predictor of poor clinical outcomes [10]. However, obesity and changes of hydration status have been shown to limit the reliability of phase angles in everyday clinical practice [11]. Bioelectrical impedance vector analysis (BIVA) [12] plots R and Xc normalized by height as a bivariate vector. Each individual vector can be ranked according to tolerance ellipses that include 50, 75, and 95% of reference values that are stratified by sex, age, and body mass index (Figure 1). This method provides better understanding of the hydration status and cell mass, and offers an easier and unbiased interpretation of the nutritional status in comparison to phase angles [13]. Moreover, BIVA has recently shown excellent sensitivity for the diagnosis of malnutrition in acutely hospitalized patients with fluid overload as compared to the classical ESPEN (European Society for Clinical Nutrition and Metabolism) criteria [14]. Therefore, the aim of our study was to assess, by BIVA, the prevalence of malnutrition in hospitalized patients with COVID-19 pneumonia, and to evaluate its relationships with the severity and outcome of the disease.

## 2. Materials and Methods

### 2.1. Study Design

This prospective, observational study was conducted at the University Hospital-University of Udine, a quaternary acute care regional hospital serving an area of more than 700,000 people in the northeast of Italy. We examined the BIVA data together with clinical, laboratory (sodium, potassium, chloride, C-reactive protein, interleuchin-6, proadrenomedullin, procalcitonin, D-dimer, creatinine, blood urea nitrogen, aspartate aminotransferase, alanine aminotransferase, bilirubin, lactic dehydrogenase, troponin, creatinkinase, albumin, uric acid, total, LDL and HDL cholesterol, triglycerides, fasting glucose), and radiological characteristics and 60-day outcomes of 150 consecutive COVID-19 patients who were hospitalized from 1 January to 28 February 2021.

### 2.2. Study Population

All patients had presented with fever and/or respiratory symptoms at specific COVID-19 points of care that were set up in the Infectious Disease and the Emergency Units of the hospital. Diagnosis of COVID-19 infection was established for confirmed (patients with pneumonia and a positive nucleic acid amplification test for SARS-CoV-2 in respiratory tract specimens) and suspected (patients with pneumonia with typical laboratory findings and/or lung CT scan imaging and/or positive serology, but negative SARS-CoV-2 NAAT) cases.

Patients’ data were collected from the University Hospital electronic database. The database is continuously and timely updated, and contains all the data of patients registered to the hospital, including patients’ demographic, anthropometric, and clinical variables, laboratory test results, historical and current diagnoses, together with history of previous hospitalizations, and historical and current medication list. Clinical severity of COVID-19 disease was assessed at admission according to the WHO classification [15]. A CT scan was performed on hospital admission in patients with severe symptoms, and within 72 h from admission in patients with mild symptoms.

Identification of cases with the COVID-19 virus was based on the detection of unique sequences of virus RNA by nucleic acid amplification tests, such as real-time PCR (RT-PCR), with confirmation by nucleic acid sequencing on respiratory samples. Genes investigated were the E gene for screening, and then, the RdRp and N genes of SARS-CoV-2 for confirmation. The viral RNA was extracted by using automated RNA extraction with the ELITe InGenius^®^ SP200 System (ELITechGroup, Puteaux, France), and RT-qPCR was performed using a LightMix^®^ Modular SARS and Wuhan CoV E-gene kit on a LightCycler^®^ 480 II instrument (Roche, Basel, Swizerland). The specimens were considered positive if the cycle threshold (Ct) value for at least one of the three genes was ≤36. The RT-PCR was conducted as recommended by the World Health Organization for COVID-19 clinical management and outbreak control purposes [16].

### 2.3. Nutritional Status: Bioelectrical Impedance Vector Analysis (BIVA)

BIVA was performed in hospitalized COVID-19 patients in a non-intensive ward setting. Height and weight were measured on admission using a digital balance with an altimeter, with patients wearing slight clothes. BIVA analysis was conducted within 36 h from admission, using a standardized method [17], when body temperature was lower than 37.5 °C. Pregnant women, patients with overt anasarca, pacemakers, arthroplasty, or active ECG monitoring were excluded because these were contraindications for BIVA. Also, patients with limb amputations or any other situation that prevented the placement of the electrodes were excluded, as well as patients in palliative care or who were unable to collaborate with the examination. BIVA was performed with the fixed frequency device, SECA^®^ (model mBCA 525; Seca gmbh & Co, Hamburg, Germany), using a tetrapolar method. Tetrapolar hand to foot measurements were obtained in subjects who were supine for 15 min. To minimize gap impedance, and in accordance with standard tetrapolar electrode placement, gel-filled electrodes were placed on the dorsal surface of the right hand and foot, respectively proximal to the metacarpal and metatarsal phalangeal joints. Measurements of resistance (R), reactance (Xc), and phase angle (PA) were obtained. Plots of the group impedance vectors were obtained from individual values of R and Xc that were transformed to Z (R) and Z (Xc) based upon the reference intervals of the adult Italian population [18]. Plots allow definition of tolerance ellipses (50th/75th/95th percentile) stratified for sex, age, and BMI from reference intervals of the older-adult Italian population [18]. Patients were classified as malnourished by the BIVA when the tip of the impedance vector was within the lower right quadrant outside the 75th percentile of the tolerance ellipse along the horizontal axis (Figure 1). Patients were followed up for 60 days from the day of hospital admission. Outcome measures were the need for invasive mechanical ventilation, 60-day mortality, and the duration of hospital stay of survivors. Outcome measures were compared in patients with COVID-19 pneumonia with or without evidence of malnutrition. The study complied with the Declaration of Helsinki, and received approval from the Regional Ethics Committee. All patients gave their written informed consent to the study.

### 2.4. Statistical Analysis

Values of normally distributed variables are expressed as mean ± standard deviation. Normality of distribution was assessed by the Kolmogorov–Smirnov test, and variables with skewed distribution were analyzed with the Mann–Whitney test. Pearson’s chi-square test was used to compare frequency distributions. A log-rank test was used to perform the survival analysis in patients with COVID-19 pneumonia with or without malnutrition. The Cox proportional hazard ratio was used to estimate the risk of mortality in malnourished patients. Patients with missing data were excluded from analyses. Data analyses were done using XL-STAT 2020 (Addinsoft, NY, USA).

## 3. Results

The demographic and clinical characteristics of enrolled COVID-19 patients are shown in Table 1. Patients’ median age was 69 years (range: 33–93; interquartile range, IQR: 58–78) and patients were prevalently male. The age range in men was 33 to 93 (IQR: 57–76), and in women, 47 to 93 (IQR: 62–81). History of hypertension, cardiovascular disease, diabetes, and dyslipidemia was present in substantial proportions. The average number of comorbidities (cardiovascular events, chronic kidney disease, liver disease, cancer, hypertension, and diabetes) was significantly higher with the growing age: 1.8 in patients with an age from 33 to 52 years; 2.6 in patients with an age from 53 to 72; and 3.9 in patients with an age from 73 to 93 (*p* < 0.001).

Patients were classified as malnourished by the BIVA when the tip of the impedance vector was within the lower right quadrant (low cellular mass) outside the 75th percentile of the tolerance ellipse along the horizontal axis, as shown in the example reported in Figure 1. BIVA revealed that 24.6% of patients had malnutrition. Examination of body composition (Table 2) revealed that COVID-19 patients with malnutrition had significantly lower fat free mass, skeletal muscle mass, and visceral adipose tissue than patients without malnutrition. Moreover, patients with malnutrition had lower total body water (TBW) and extracellular water (EBW), but an increased EBW/TBW ratio. Malnourished patients were significantly older, and had lower BMI, waist circumference, and serum albumin level, and higher serum pro-adrenomedullin and D-dimer than patients without malnutrition (Table 1). The prevalence of comorbidities was comparable between patients with or without malnutrition, except for previous history of cardiovascular diseases, which was more frequent in the former group. No significant differences were observed in the remaining variables. Table 3 summarizes the clinical severity scores of COVID-19 pneumonia according to WHO, and the severity of lung involvement as detected by CT scans. No significant differences were observed between patients with or without malnutrition.

### Outcome Variables

Table 3 reports the number of patients who required invasive mechanical ventilation and of those who died during the 60-day follow-up, together with duration of the hospital stay before discharge in those who survived COVID-19. Malnourished patients with COVID-19 pneumonia needed invasive mechanical ventilation more frequently than patients without evidence of malnutrition on BIVA. During follow-up, mortality in malnourished patients was 4- to 5-fold that of patients with normal nutritional status. Duration of hospitalization was longer by an average of 5 days in patients with malnutrition than in patients without malnutrition. In a survival analysis, invasive mechanical ventilation free survival (Figure 2) and overall survival (Figure 3) were significantly longer in COVID-19 patients with no evidence of malnutrition than in COVID-19 patients with evidence of malnutrition on BIVA (log-rank 7.887, *p* = 0.050 and log-rank 17.886, *p* < 0.001, respectively). With the use of the Cox proportional ratio (Table 4 and Table 5), malnutrition was associated with an increased risk of invasive mechanical ventilation (hazard ratio 4.375, *p* = 0.004) and death (hazard ratio 4.478, *p* = 0.004), even after adjustment for major confounders such as age, sex, and BMI.

## 4. Discussion

This prospective study was designed with the aim to assess the prevalence of malnutrition in hospitalized patients with COVID-19 pneumonia, and the relevance of malnutrition for disease severity and short-term survival. Malnutrition was detected by use of BIVA in one fourth of patients hospitalized for COVID-19, with a prevalence that is consistent with what has been previously reported, with the use of the same methodology in patients hospitalized for other reasons [10]. Although many factors, including age, ethnicity, geographical area, socio-economic level, and, obviously, the specific clinical context, could concur to determine the prevalence of malnutrition, we might reasonably state that the prevalence of malnutrition observed in this study reflects that which is ordinarily found in medical departments.

To date, only few studies have investigated the body composition of hospitalized patients with COVID-19, and its prognostic relevance. One study did not report any association between body composition, disease severity, and outcomes in 54 patients (median age 67, IQR 64–71) hospitalized for COVID-19 in a general ward or in intensive care unit, however, a lower phase angle did increase the odds of severe COVID-19 [19]. In another recent study, it has been reported that a low phase angle (<3.95°) detected by use of BIVA is a significant predictor of mortality in hospitalized COVID-19 (median age 69, IQR 59–71) patients, independently of age, sex, and BMI [20]. Conversely, in a retrospective study conducted in 90 hospitalized patients (mean age 65 ± 14 years) with COVID-19, a low phase angle was not associated with longer hospitalization or composite of death and intensive care unit admission [21]. In our hands, the phase angle was associated with an increased risk of death at 60 days in a univariate model, but statistical significance was lost after correction for age and sex. The phase angle is directly correlated with lean body mass (LBM) and body cellular mass (BCM), but inversely related to the ratio of extracellular (ECW) to intracellular water (ICW) in healthy adults [13]. Disease-related malnutrition is characterized by an early shift of fluids from ICW to ECW space, with increased ECW/ICW and a concomitant decrease in BCM both lowering phase angle. These disease-related changes in body fluid distribution (found even in our study) affect phase angle measurements [22], and this is why the applicability of this parameter in critically ill patients is of limited value [23], and might explain, at least in part, an inconsistency in findings.

The BIVA method provides additional information on hydration and cell mass integrity that allows more accurate distinction between patients with malnutrition associated with a change of the hydration status [24], and patients with critical illness [25]. This aspect holds specific relevance for the patients included in our study who had frequent fever at initial presentation, and were often treated with antipyretics. In our patients with COVID-19 pneumonia, malnutrition was defined by measurement of resistance (R) and reactance (Xc), which were used to obtain the impedance vector. Malnutrition was associated with significantly higher rates of the use of invasive mechanical ventilation, and, most importantly, with higher short-term mortality. Also, malnourished patients hospitalized with COVID-19 who survived the disease had a longer duration of hospitalization. To our knowledge, this is the first prospective study to demonstrate an increased risk of invasive mechanical ventilation and death in malnourished patients with COVID-19, supporting the possible use of BIVA as a prognostic tool in this setting. Consistent with our present findings, a retrospective study conducted in China in severe or critically ill patients reported that severe malnutrition diagnosed by the Nutritional Risk Screening 2002 (NRS-2002) was associated with the highest mortality rate [26]. Similar findings were found even in European populations, where NRS-2002 score < 3 at the time of ICU admission increased the risk of in-hospital death [27]. However, in a French study on 114 patients hospitalized with COVID-19, the nutritional status, as defined by the Global Leadership Initiative on Malnutrition (GLIM), was not associated with worse outcomes. [28]

Increased mortality in malnourished patients could be explained by several factors. Similar to any other physiological system, the immune system is affected by the nutritional status. In the case of malnutrition, the lack of protective immunity can be easily traced back to the developmental defects associated with inadequate nutrients, and the lack of nutritional signals, such as leptin, that are critical for fueling immune cell proliferation and function [29]. Even micronutrients defects have been associated with a decreased immune response and a higher susceptibility to respiratory infections [30]. For this reason, malnutrition has a devastating effect on the ability of the immune system to mount a successful immune response to infection, and, therefore, is associated with poor outcomes. Consistently, in a recent study on Chinese COVID-19 patients, malnutrition was associated with hyperinflammation and immunosuppression, which could eventually lead to worse outcomes [31]. In support of this hypothesis, the malnourished patients included in our study had increased serum pro-adrenomedullin, recently identified as a good prognostic inflammatory marker in COVID-19 patients [32].

Major strengths of our study are the prospective design and the availability of a large amount of clinical, biochemical, and imaging data that permit accurate characterization of COVID-19 patients. There are also some important limitations. First, we included only patients who were hospitalized in a general ward setting, and, therefore, it is not possible to extrapolate findings on the prevalence of malnutrition and its prognostic values to other settings of care, such as home or intensive care units. Second, we could enroll only patients who were eligible for BIVA, thus excluding some patients treated with medical devices that could be used in critically ill COVID-19 patients.

## 5. Conclusions

Malnutrition, as defined by the use of BIVA, which detects decreased cellular mass, is frequently found in COVID-19 patients who are hospitalized in non-intensive medical units. In these patients, malnutrition hits more severely patients older than 55 years, and is independently associated with a more frequent need of invasive mechanical ventilation, and higher mortality in the short-term. These findings support the use of BIVA as a useful prognostic marker in hospitalized patients with COVID-19, and underline the importance of early nutritional screening (at least with simpler diagnostic tools, such as NRS-2002), in particular, in elderly patients. Appropriately designed intervention studies will be needed to test the possible benefits of early nutritional support on the survival of patients hospitalized with COVID-19.

## Figures and Tables

**Figure 1 nutrients-13-04085-f001:**
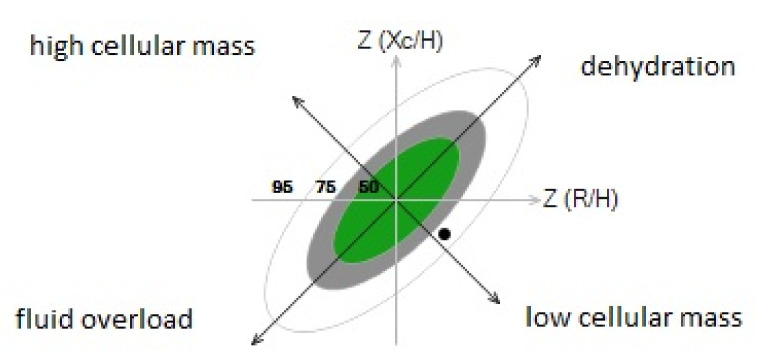
R/Xc graph provided by bioimpedance vector analysis according to the method of Piccoli et al. [12]. Patients were classified as malnourished when the tip of the impedance vector was within the lower right quadrant outside the 75th percentile of the tolerance ellipse along the horizontal axis (black dot in the figure).

**Figure 2 nutrients-13-04085-f002:**
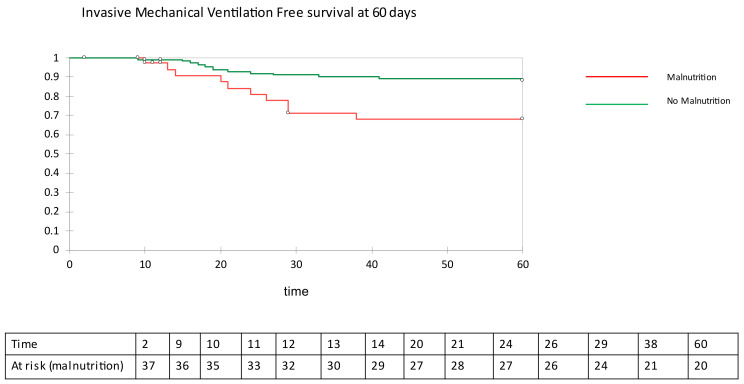
Invasive mechanical ventilation free survival at 60 days.

**Figure 3 nutrients-13-04085-f003:**
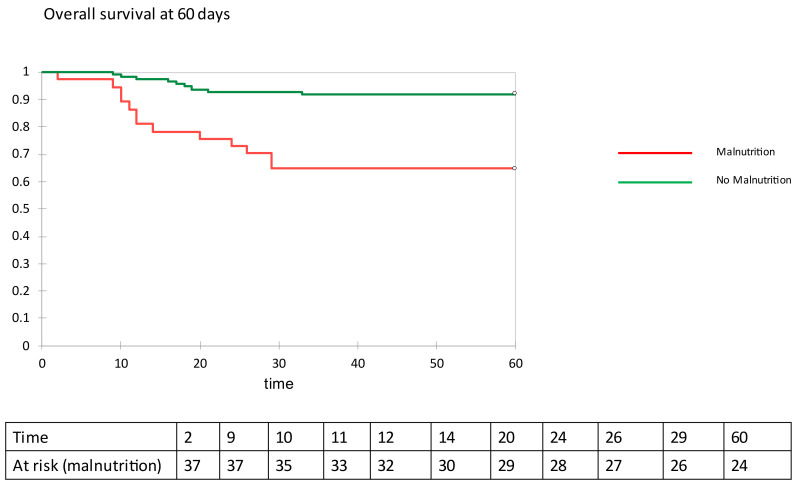
Overall survival at 60 days.

**Table 1 nutrients-13-04085-t001:** Baseline clinical characteristics and laboratory data.

Variable	All Patients (*n* = 150)	No Malnutrition (*n* = 113)	Malnutrition (*n* = 37)	*p*-Value
Age (years) Median and IQR	69 (58–78)	66 (57–75)	77 (71–83)	<0.001
Male Sex (%)	68.7% (103)	69.1% (78)	67.6% (25)	0.860
BMI (kg/m^2^)	28.3 ± 4.9	29.3 ± 5.1	25.2 ± 3.4	<0.001
Waist Circ (cm)	102 ± 16	104 ± 14	96 ± 11	0.001
CRP (mg/dL)	90.3 ± 71.4	86.7 ± 70.4	101.8 ± 74.4	0.145
Current Smoker (%)	3.4	2.7	3.6	0.237
Dyslipidemia (%)	26.2	26.8	24.3	0.957
Hypertension (%)	55.1	56.3	51.4	0.604
Diabetes (%)	28.8	27.7	32.4	0.570
COPD (%)	3.3	2.7	5.4	0.508
CV disease (%)	25.5	21.4	37.8	0.025
CKD (%)	12.6	10.7	16.6	0.126
Malignancies (%)	10.7	8	18	0.064
Liver disease (%)	6.7	6.3	8.3	0.187
PCT (ng/mL)	0.34 + 1.35	0.89 + 0.11	0.16 + 0.18	0.120
IL-6 (ng/mL)	66.5 ± 83.7	63.8 ± 63.9	72.2 ± 117.5	0.440
Pro-ADM (ng/mL)	1.22 ± 0.93	1.09 ± 0.45	1.61 ± 1.61	0.012
D-Dimer (ng/mL)	1584 ± 4025	1344.7 ± 3951.7	2309 ± 4217.2	0.025
BUN (mg/dL)	24.7 ± 10.1	22.6 ± 10.8	25.4 ± 10.8	0.712
Creatinine (mg/dL)	1.2 ± 1.8	1.1 ± 0,4	1.5 ± 2.3	0.667
Sodium (mEq/L)	138.2 ± 4.1	138.5 ± 3.7	137.2 ± 4.6	0.073
Potassium (mEq/L)	4.13 ± 0.5	4.12 ± 0.5	4.15 ± 0.5	0.853
Chloride (mEq/L)	100.4 ± 4.8	100.2 ± 4.2	101.2 ± 6.4	0.699
AST (UI/L)	41 ± 26.4	41.1 ± 26	37.6 ± 21.3	0.705
ALT (UI/L)	42.3 ± 36.3	44.7 ± 39.3	34.4 ± 23.9	0.156
Bilirubin (mg/dL)	0.568 ± 0.29	0.59 ± 0.31	0.48 ± 0.19	0.081
LDH (UI/L)	651.8 ± 241.9	640.4 ± 240.5	686.9 ± 246.5	0.328
CPK (UI/L)	155.3 ± 218.4	170.1 ± 230.2	110.2 ± 172.8	0.102
Troponin (ng/mL)	17.4 ± 16.2	16.7 ± 15.1	19.6 ± 9	0.133
Albumin (g/L)	34.4 ± 4.1	34.8 ± 4.1	33.1 ± 3.3	0.020
Uric Acid (mg/dL)	4.7 ± 1.8	4.8 ± 1.6	4.6 ± 2.2	0.457
Cholesterol (mg/dL)	150.2 ± 36.7	157.5 ± 30.7	147.6 ± 38.3	0.328
HDL (mg/dL)	33.4 ± 9.5	34.9 ± 8.9	32.8 ± 9.7	0.207
LDL (mg/dL)	84.8 ± 30.9	90.9 ± 25.8	82.7 ± 32.3	0.442
TG (mg/dL)	165.2 ± 58.8	165.9 ± 60.9	163.3 ± 53.4	0.729
Glucose (mg/dL)	130.9 ± 58.1	130.4 ± 57.5	132.4 ± 60.6	0.849
HbA1c (DCCT)	6.6 ± 1.1	6.5 ± 0.9	6.6 ± 1.2	0.950

IQR, interquartile range; COPD, chronic obstructive pulmonary disease; CKD, chronic kidney disease; CRP, C-reactive protein; PCT, procalcitonin; Pro-ADM, proadrenomedullin; BUN, blood urea nitrogen; AST, aspartate aminotransferase; ALT, alanine aminotransferase; LDH, lactic dehydrogenase; CPK, creatin-phosphokinase; HDL, high density lipoprotein; LDL, low density lipoprotein.

**Table 2 nutrients-13-04085-t002:** Body composition data obtained by bioimpedance analysis in the study patients.

Variable	All Patients (*n* = 150)	No Malnutrition (*n* = 113)	Malnutrition (*n* = 37)	*p*-Value
Fat mass (kg/m^2^)	28.8 ± 10.1	28.2 ± 10.6	30.5 ± 8.3	0.293
Visceral adipose tissue (lt)	3.4 ± 2.1	3.7 ± 2.2	2.8 ± 1.5	0.037
Fat free mass (kg/m^2^)	59.1 ± 13.3	62.1 ± 13.1	49.9 ± 9.3	<0.001
Skeletal muscle mass (kg/m^2^)	27.1 ± 8.4	29.1 ± 8.2	20.9 ± 5.7	<0.001
TBW	44.6 ± 10.1	46.3 ± 10	37.0 ± 6.6	<0.001
EBW	12.2 ± 3.9	20.4 ± 3.9	17.3 ± 2.5	<0.001
EBW/TBW	45.1 ± 3.3	44.4 ± 3.2	47 ± 2.7	<0.001
Phase angle (°)	5.5 ± 1.5	5.9 ± 1.5	4.5 ± 0.7	<0.001

TBW, total body water; EBW, extracellular body water.

**Table 3 nutrients-13-04085-t003:** Clinical severity of COVID 19 pneumonia (WHO stage), lung CT scan involvement, and outcome variables in the study patients. IMV: invasive mechanical ventilation.

Variable	All Patients (*n* = 150)	No Malnutrition (*n* = 113)	Malnutrition (*n* = 37)	*p*-Value
WHO stage				
0	13.3% (20)	14.2% (16)	10.8% (4)	0.738
1	33.3% (50)	36.3% (41)	24.3% (9)	0.229
2	38.6% (58)	37.1% (42)	43.2% (16)	0.562
3	14.8% (22)	12.4% (14)	21.7% (8)	0.185
Lung CT Extension				
<25%	9.6% (15)	9.3% (11)	10.8% (4)	0.986
25–50%	37.5% (56)	42.7% (45)	24.1% (9)	0.114
50–75%	49% (73)	45.3% (51)	58.5% (22)	0.227
>75%	3.8% (6)	2.7% (4)	6.4% (2)	0.310
Mortality (60 days) (%)	14.7% (22)	7.9% (9)	35.1% (13)	0.001
IMV (%)	15.3% (23)	11.5% (13)	27.1% (10)	0.023
Hospital stay (days)	14.5 ± 15.1	13.2 ± 14.8	18.2 ± 15.7	0.003

IMV, invasive mechanical ventilation.

**Table 4 nutrients-13-04085-t004:** Cox proportional hazard ratio for need of mechanical ventilation at 60 days. Crude and correct for covariates.

Variable	Value	Standard Error	Wald Chi-Square	Pr > Chi^2^	Hazard Ratio	Lower Bound (95%)	Upper Bound (95%)
BIVA Malnutrition (crude)	1.122	0.421	7.086	0.008	3.069	1.344	7.009
BIVA Malnutrition (adjusted) *	1.476	0.513	8.275	0.004	4.375	1.601	11.959
Phase Angle value (adjusted) *	0.007	0.176	0.002	0.967	1.007	0.714	1.422

* Adjusted for age, sex, and BMI.

**Table 5 nutrients-13-04085-t005:** Cox proportional hazard ratio for death at 60 days. Crude and correct for covariates.

Variable	Value	Standard Error	Wald Chi-Square	Pr > Chi^2^	Hazard Ratio	Lower Bound (95%	Upper Bound (95%)
BIVA Malnutrition (crude)	1.641	0.434	14.287	0.000	5.159	2.203	12.081
BIVA Malnutrition (adjused) *	1.498	0.516	8.421	0.004	4.474	1.626	12.306
Phase Angle value (adjused) *	0.081	0.153	0.280	0.597	1.084	0.803	1.463

* Adjusted for age, sex, and BMI.

## Data Availability

Data would be avalaible (if required) only after the authorization of local ethics committee.

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
