# Peer review of "Prognostic Role of Malnutrition Diagnosed by Bioelectrical Impedance Vector Analysis in Older Adults Hospitalized with COVID-19 Pneumonia: A Prospective Study"

_nutrients, 2021, doi:10.3390/nu13114085_

Round 1

Reviewer 1 Report

  1. I congratulate the authors on the work done on this relevant subject.

    Malnutrition in critical care is highly prevalent and well documented to have adverse implications on morbidity and mortality. However, nutrition is an important element of care. The nutritional assessment and the early nutritional care management of COVID-19 patients must be integrated into the overall therapeutic strategy.

    I would like to contribute with some comments, suggestions and questions to the authors:

    1. The information about affiliation is incomplete - please look to Nutrients Microsoft Word template and other manuscripts in Nutrients. Please delate word Affiliation 1 and use correct form e.g. 1. Department of Precision Medicine, University of Campania “Luigi Vanvitelli”, 80138 Naples, Italy; [email protected]
    2. Line 100-102 please describe the method not send us to other paper. Manuscript should be complete. 
    3. Please divided section 2. Material and Methods - it will be more readable. e.g.
      2.1. Study design
      2.2. Study Population
      2.3. Nutritional Status - bioelectrical impedance vector analysis etc. 

    I believe that this part will be more readable.

    1. References should be numbered in order of appearance and indicated by a numeral or numerals in square brackets—e.g., [1] or [2,3], or [4–6]. 
    2. In all table you should add (under the table) abbreviation. 
    3. Table 1-Please add unit in D-Dimer
    4. TABLES - sometimes you use bracket in units, sometimes no (e.g Troponin). Please add bracket in missed places. 
    5. TABELES: Sometimes you use full name eg. Total body water % (TBW) and sometimes shortcut. In all table you should use only shortcut and add abbreviation under the table.
    6.  line 217-223 The risk of malnutrition at the time of ICU admission also increases the risk of in-hospital death in central Europe eg. Czapla, M.; Juárez-Vela, R.; Gea-Caballero, V.; ZieliÅ„ski, S.; ZieliÅ„ska, M. The Association between Nutritional Status and In-Hospital Mortality of COVID-19 in Critically-Ill Patients in the ICU. Nutrients 202113, 3302. https://doi.org/10.3390/nu13103302 
    7. The limitations as stated by the authors are reasonable and can be appreciated and considered.
    8. Overall, the authors have analyzed the available data to a reasonable conclusion.

Reviewer 2 Report

Dear Authors,

the aim of the study is very interesting and worth developing. However, I have some technical and substantive comments.

Technical comments

-In the text, reference numbers should be placed in square brackets [ ], please correct.

-References should be written in the same font as the manuscript - Palatino Linotype - see instructions for authors.

-The list of references should be written in accordance with the instructions for authors - please read and correct.

Substantive comments

Introduction

In the introduction, the information on the importance of the phase angle in the nutritional status assessment should be supplemented.

Materials and Methods

  1. Line 72 - ‘All patients had presented with fever’ - Fever affects hydration. Body temperature is also related to the amount of body fat.

        Has fever been taken into account when measuring body composition? Shouldn't the results be    corrected for body temperature? How can you comment on this?

        Has the use of antipyretic drugs been considered?

  1. In what age range were the patients - separately women and men? It would be nice to mention it in this chapter as it is important for the interpretation of the results.
  2. Line 82 – Please list, in parentheses, all laboratory tests that have been pooled.
  3. Please explain what was the criterion for dividing patients into malnutrition and non-malnutrition?

It has not been made clear.

Results

  1. Comments for lines:137-141 - Patients were over 50 years old and mostly over 60 years old. At this age, body composition changes to reduce lean body mass. Perhaps malnutrition should be classified differently here. BMI values can also be categorized differently for the age of 65 and over. As a result, 'malnutrition' can be closely correlated with age and should not be a criterion in itself.

       Please refer to these comments in the results discussion.

  1. Lines 143-145. ‘Prevalence of comorbidities was comparable in patients with or without malnutrition’, - but maybe it was age related?

Discussion

  1. Lines 217-223 - here is information about the results of other studies. Did these studies use the same criteria for assessing malnutrition as the Authors? Were the patients of a similar age? This requires some explanation.

In the discussion of the results, it is necessary to confront the results of other studies, whether the age of the patients was similar or different. If different, what were the differences in the parameters?

Conclusion

Conclusion should be restricted to patients over 55 years of age

General remark

In my opinion, age is a very important predictor of what can be considered 'malnutrition' but not an abuse condition. Perhaps the condition that was diagnosed as 'malnutrition' in these studies was not just due to improper food intake, but was due to age-related changes in body composition.

Therefore, I believe that this should be marked in the manuscript title:

 Prognostic role of bioelectrical impedance vector analysis diagnosed malnutrition in hospitalized patients over 55 years old with COVID-19 pneumonia: a prospective study

Sincerely Yours

Round 2

Reviewer 1 Report

I reccomend to accept 

Reviewer 2 Report

Dear Authors,
thank you very much for the detailed explanations and additions to the manuscript. Now it gives a full view of methods and interpretations.

Best regards